# DETECTING SHORTCUTS USING MUTUAL INFORMATION

## ABSTRACT

The failure of deep neural networks to generalize to out-of-distribution (OOD) data is a well-known problem that raises concerns about the deployment of trained networks in safety-critical domains such as healthcare and autonomous vehicles. We study a particular kind of distribution shift — *shortcuts* or *spurious correlations* in the training data. These correlations are not present in real-world test data, so there is a performance drop due to distribution shift, also referred to as shortcut learning. Shortcut learning is often only exposed when models are evaluated in carefully controlled experimental settings, posing a serious dilemma for AI practitioners to properly assess the effectiveness of a trained model for real-world applications. In this work, we try to understand shortcut learning using information-theoretic tools and propose to use the mutual information (MI) between the learned representation and the input space as a domain-agnostic metric for detecting shortcuts in the training datasets. For studying the training dynamics of shortcut learning, we develop a Neural Tangent Kernel (NTK) based framework, which can be used to detect shortcuts and spurious correlations in the training data without requiring class labels of the test data. We empirically demonstrate on multiple datasets, such as MNIST, CelebA, NICO, Waterbirds, and BenchMD, that MI can effectively detect shortcuts. We benchmark against multiple OOD detection baselines to show that OOD detectors cannot detect shortcuts, and our method can be used in complementary with OOD detectors to identify all types of distribution shifts in the datasets, including shortcuts. Codes and datasets are available on our **anonymous repository**.

## 1 INTRODUCTION

Our understanding of 'how' and 'what' neural networks learn is limited, which raises concerns about the deployment of neural networks in safety-critical domains. Despite achieving state-of-the-art performance on benchmark datasets, neural networks may fail to generalize in real-world settings or for out-of-distribution data (Koh et al., 2021). For example, models trained for cancer detection may not generalize on data from a new hospital (Castro et al., 2020; Perone et al., 2018; AlBadawy et al., 2018), and self-driving cars may not generalize to new lighting conditions or object poses (Alcorn et al., 2018; Dai & Van Gool, 2018). One reason why models may fail in real-world settings may be attributed to learning *shortcuts* (Geirhos et al., 2020) from the training data. A *shortcut* is a type of distribution shift where spurious correlations exist only within the dataset used for training and evaluating the model, resulting in the learning of non-intended or easy-to-learn discriminatory features which work well on the training and test dataset but not on out-of-distribution real-world datasets (Wiles et al., 2022; Geirhos et al., 2020). Shortcuts can arise due to dataset biases or the model using 'trivial' or unintended features like high-frequency noise patterns or the image background in a classification task. For example, deep learning model trained to detect COVID-19 from chest radiographs can rely on confounding factors (shortcuts) rather than medical pathology (DeGrave et al., 2021). Zech et al. (2018) studied the failure of models trained for classifying pneumonia from X-rays and found that the models had learned to identify particular hospitals by detecting hospital-specific tokens. While our understanding of shortcuts and how they arise is still developing, a helpful tool to practitioners deploying machine learning models in safety-critical domains with a high cost of failure would be to detect shortcuts in the training data. Although the phenomenon of shortcut learning is widely known, there is no effective method available to detect shortcut learning. Interpretable machine-learning methods such as feature attribution, Grad-CAM (Selvaraju et al., 2017), and

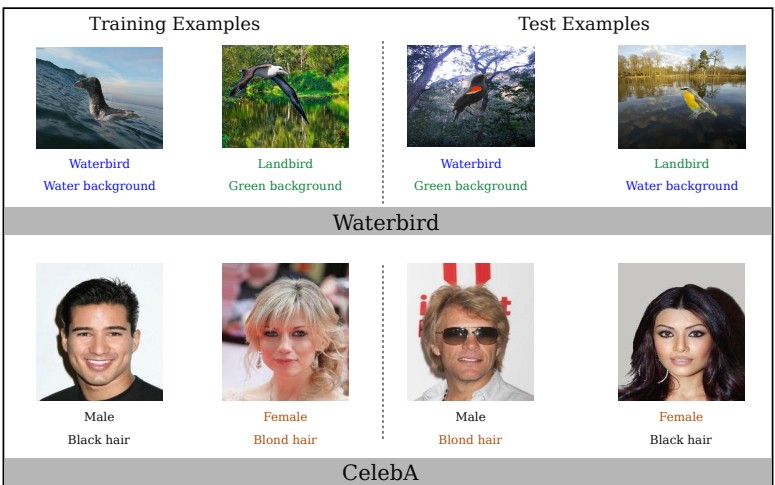

Figure 1: Representative images from Waterbird and CelebA dataset.Waterbirds have water background and landbirds have green background; CelebA dataset is sampled such that in the training data, images of males have black hair and females have blonde hair.

LIME (Ribeiro et al., 2016a) have been used to understand a model's dependency on spurious correlations. However, it has been shown that such post-hoc explanations are ineffective (Adebayo et al., 2021; Alqaraawi et al., 2020; Chu et al., 2020; Kindermans et al., 2017; Adebayo et al., 2020).

In this work, we show that shortcut learning arising due to dataset biases can be understood using the information bottleneck framework (Tishby et al., 1999; Tishby & Zaslavsky, 2015) by using the mutual information (MI) between the inputs and the learned representation. We use the neural tangent kernel (NTK) (Jacot et al., 2018) to study the training evolution of shortcut learning. We design experiments using synthetic and complex real-world data to demonstrate the relationship between (MI) and shortcut learning and show that MI can be used as a metric for domain-agnostic detection of shortcuts. We find that compression, as measured by MI, is associated with the tendency to learn shortcuts. We benchmark against multiple OOD detection methods and experimentally demonstrate, using multiple datasets, that our NTK-based framework can detect shortcuts in the training data. Thus, our method can be used in complmentary to existing OOD detectors as a domain-agnostic metric to detect shortcuts and diversify the training data before deploying a model in safety-critical domains.

## 2 BACKGROUND

**Shortcut learning.** Wiles et al. (2022) defined shortcuts or spurious correlations as a type of *distribution shift* such that two or more attributes are correlated at training time, but not for the test data, where they are independent. In a more general sense, shortcuts are a kind of decision rule that can be exploited in the absence of distribution shift (i.e. on standard benchmarks) but fail to transfer to more challenging and diverse testing conditions, such as real-world datasets (Geirhos et al., 2020). For example, a model trained to classify cows can use grassy landscapes as a shortcut. Using shortcuts as a decision rule, the model can perform well on data having the same shortcut, but the model fails to generalize on images that do not contain the shortcut, for example, cows on a beach.

Shortcuts can arise due to the following reasons (Geirhos et al., 2020):

1. Network architecture: The inductive bias of the model induces a bias on the learned features. A network is more robust if its architecture is more aligned with the target function than the shortcut feature or noise (Li et al., 2021). For example, convolutional Neural Networks are biased toward learning texture; MLPs are biased toward the spatial location of features.

2. Optimization: Optimization methods also induce bias towards different features. Stochastic gradient descent is biased toward learning simple functions (Wu et al., 2017; De Palma et al., 2018). It has been shown that the learning rate also affects the complexity of the learned function (Li et al., 2019).

3. Dataset biases: Bias in the training dataset can allow the model to learn shortcuts instead of meaningful representations. For instance, in the cow example above, shortcuts arise due to the sampling bias of the dataset. *This work focuses on shortcut learning arising due to the biases and spurious correlations in the training data.*

**Information bottleneck method.** The information bottleneck (IB) can be viewed as a rate-distortion problem cast entirely in terms of mutual information (MI) — denoted "$I(\cdot;\cdot)$" (Tishby et al., 1999). A *distortion* function measures how well a relevant variable $Y$ can be predicted from another variable $Z$, where $Z$ is usually a compressed representation of the input $X$. The *rate* refers to the complexity of $Z$, which is less than or equal to $X$. IB is a general method for data compression but has been advanced by Tishby & Zaslavsky (2015) as an explanatory tool predictive of learning and generalization of neural networks (NNs). It has been suggested that NNs trained by SGD may learn compressed representations $Z$ of their input, making them insensitive to data idiosyncrasies, yet maintain sufficient *relevant* information for predicting the output $Y$ (e.g. class labels) (Tishby & Zaslavsky, 2015; Shwartz-Ziv & Tishby, 2017). This trade-off between compression and preserving task-relevant information is optimized by the notion of "minimal sufficient statistics" (Cover & Thomas, 1991). The IB view suggests that NNs trained by cross-entropy loss may implicitly minimize the following Lagrangian:

$$\min I(X;Z) - \beta I(Z;Y), \tag{1}$$

enabling them to implement minimal sufficient statistics for different $\beta$-constraints on the error.[1]

**Neural tangent kernel** Distribution-free estimation of MI for high-dimensional data is challenging and intractable. One workaround to this problem is using an infinite ensemble of infinite-width neural networks which confer tractable bounds on MI (Shwartz-Ziv & Alemi, 2020; Galloway et al., 2023). The Neural Tangent Kernel (NTK) (Jacot et al., 2018) is a kernel that describes the training evolution of infinite-width neural networks by gradient descent, thus allowing the systematic study of neural networks using tools from kernel methods. Infinite-width neural networks behave as linear functions, and their training evolution can be fully described by the NTK. This allows tractable computation of MI between the representation $Z$ and the targets $Y$: $I(Z;Y)$; and the MI between $Z$ and the inputs $X$ during training: $I(Z;X)$.

## 3 METHODOLOGY

**Notation.** Let $(X_{tr}, Y_{tr})$ denote the samples from the IID training dataset and $X_{test}$ denote the unlabelled samples from OOD test data. Let $v = \{v_s, v_t\}$ represent the underlying latent factors for generating input data, where $v_s$ are the latent factors corresponding to the shortcut features and $v_t$ are the latent factors corresponding to the true semantic information. The spurious correlation label, denoted $s$, is generated by the latent vector $v_s$, and $y$ is the actual class label generated by $v_t$. We refer to OOD real-world data as test data unless stated otherwise, which should not be confused with the usual training, validation, testing split drawn from the same distribution.

**Definition 1.** Latent factors $v_s$ are spuriously correlated if it is correlated with the training label $Y_{tr}$, but not with the out-of-distribution real-world data. Murali et al. (2023) mathematically defined spurious correlation in terms of the joint probability distributions of the training and test data as:

$$P_{tr}(x,y,s) = P(x|s,y)P(s|y)P(y) \tag{2}$$
$$P_{test}(x,y,s) = P(x|s,y)P(s)P(y). \tag{3}$$

**Definition 2.** Let $\Gamma(X \to Y)$ be some notion of generalization of a model learning the actual class labels and $\Gamma(X \to s)$ be the notion of generalization of a model learning the spurious correlation label $s$ or the unintended decision rule. Spurious correlation $s$ is said to be a shortcut if it doesn't generalize well on datasets other than the data from the training distribution, i.e., $\Gamma(X \to Y) \gg \Gamma(X \to s)$.

**Proposition 1.** In the presence of shortcuts, models prefer to learn features corresponding to latent factors $v_s$ over $v_t$. This can be explained using the information bottleneck (IB) theory. According to the information bottleneck theory, neural networks tend to learn a compressed representation $Z$ of the input $X$. Since $v_s$ contains less information (Yang et al., 2022), shortcuts in the training data allow networks to learn a maximally compressed representation $Z$, i.e., $I(X;Z)$ is considerably reduced while allowing the model to achieve high training accuracy.

---

[1]To what extent the relationship between IB and deep learning holds in general is the subject of ongoing debate (Saxe et al., 2018; Jacobsen et al., 2018; Goldfeld et al., 2019).

**Problem Formulation.** The objective is to detect any shortcuts in a labelled training dataset $D_{tr}$ in relation to unlabeled OOD data $X_{test}$ and determine whether a model trained on $D_{tr}$ can be accurately used to classify or predict $X_{test}$.

**Proposed Method.** Our hypothesis is that MI, as measured during the evolution of a network's parameters, can be used as a domain-agnostic metric to detect shortcuts in the training dataset.

We compare the $I(X;Z)$ on the training data $X_{tr}$ and the OOD real-world data $X_{test}$. We hypothesize lower $I(X;Z)$ on the OOD real-world data indicates the presence of shortcuts in the training data. We use NTK to train the model on the training data and compute the mutual information as NTK yields a tractable distribution of the output.

**Estimating MI using NTK.** Infinitely-wide networks are linear functions of their parameters, which makes them analytically tractable (Jacot et al., 2018). An infinite-width neural network trained using mean squared loss admits a closed-form equation for output ($z$) for any time (epoch) $t$,

$$z(x,t) = z_0(x) - \Theta(x,\mathcal{X})\Theta(\mathcal{X},\mathcal{X})^{-1}(I - e^{-t\Theta(\mathcal{X},\mathcal{X})})(z_o(\mathcal{X}) - \mathcal{Y}), \tag{4}$$

where $\mathcal{X}$ is the training data and $x$ is the input.

The evolution of the output in the infinite width is determined by the Neural Tangent Kernel $\Theta$, which converges in probability to a fixed value. Since the evolution of $z(x)$ is an affine transformation of $z_o(x)$, $z(x)$ follows Gaussian distribution during the training evolution:

$$p(z|x) \sim \mathcal{N}(\mu(x,t), \sigma(x,t)), \tag{5}$$

$$\mu(x,t) = \Theta(x,\mathcal{X})\Theta(\mathcal{X},\mathcal{X})^{-1}(I - e^{-t\Theta(\mathcal{X},\mathcal{X})})\mathcal{Y}, \tag{6}$$

$$\Sigma(x,t) = \mathcal{K}(x,x) + \Theta(x,\mathcal{X})\Theta^{-1}(I - e^{-t\Theta})(\mathcal{K}\Theta^{-1}(I - e^{-t\Theta})\Theta(\mathcal{X},x) - 2\mathcal{K}(\mathcal{X},x)), \tag{7}$$

where $\Theta \equiv \Theta(\mathcal{X},\mathcal{X})$ and $\mathcal{K}$ is the Neural Network Gaussian Process (NNGP) kernel (Lee et al., 2018a). The lower bound on the $I(X;Z)$ for any time $t$ can then be calculated using the multi-sample unnormalized lower bound (Poole et al., 2019; Shwartz-Ziv & Alemi, 2020):

$$\frac{1}{N}\sum_i \log \frac{p(z_i|x_i)}{\frac{1}{N}\sum_j p(z_i|x_j)} \leq I(X;Z). \tag{8}$$

Similarly, the lower-bound of $I(Z;Y)$ can be calculated using the variational lower bound (Poole et al., 2019) in terms of the entropy $H$ and variational distribution $q(y|z)$:

$$H(Y) + \mathbb{E}[\log q(y|z)] \leq I(X;Y). \tag{9}$$

---

**Algorithm 1:** Overview of the proposed method for detecting shortcuts.

1. Given a trained model $\mathcal{F}$ on training data $(X_{tr}, Y_{tr})$ and unlabelled OOD test data $X_{test}$:

2. Train an infinite-width model with the same architecture as $\mathcal{F}$ on the training data $X_{tr}$.

3. Using NTK, compute $I(X;Z)$ on the samples from the training data $X_{tr}$ and OOD real-world data $X_{test}$

4. **if** $I(X_{test};Z) < I(X_{tr};Z)$ **then** the training dataset contains shortcuts and the model $\mathcal{F}$ cannot generalize well on the test data.

---

## 4    EXPERIMENTS AND OBSERVATIONS

**Experimental setup.** To validate our hypothesis, we design controlled datasets containing shortcuts and measure $I(X;Z)$ on the OOD test data during the training evolution using the NTK. Using insights from a previously introduced mutual information plane visualization (Shwartz-Ziv & Tishby, 2017) and mutual information profile on the training and the OOD test dataset, we can detect shortcuts in the training dataset. For each experiment, we plot $I(X;Z)$ and $I(Z;Y)$ w.r.t. time, generalization error/loss on the OOD test data, and the information plane ($I(Z;Y)$ vs. $I(X;Z)$). Note that we calculate the upper bound of $I(X;Z)$.

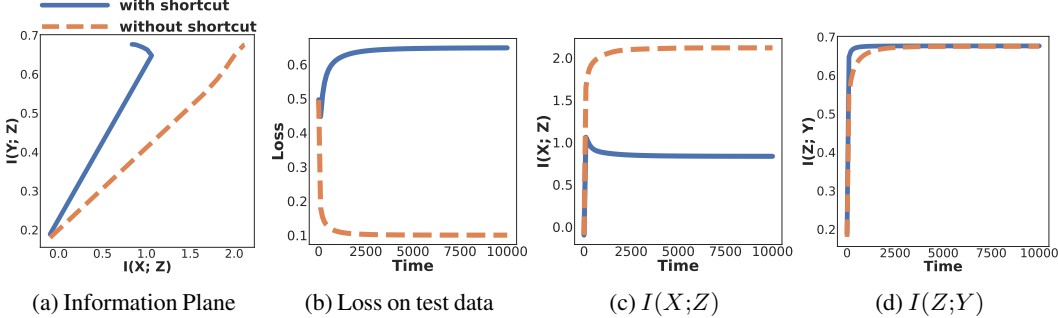

Figure 3: Comparison of training evolution with and without synthetic shortcuts on MNIST dataset. An upper bound on MI is plotted. (a) Information plane showing the $I(X;Z)$ and $I(Z;Y)$ values during the training. (b) Plot of loss value on the clean test data without shortcuts — OOD for blue line, ID for orange. (c) Plot of $I(X;Z)$ during the training evolution. (d) Plot of $I(Z;Y)$ evaluated on the training set. Animated GIF of the plot can be viewed here.

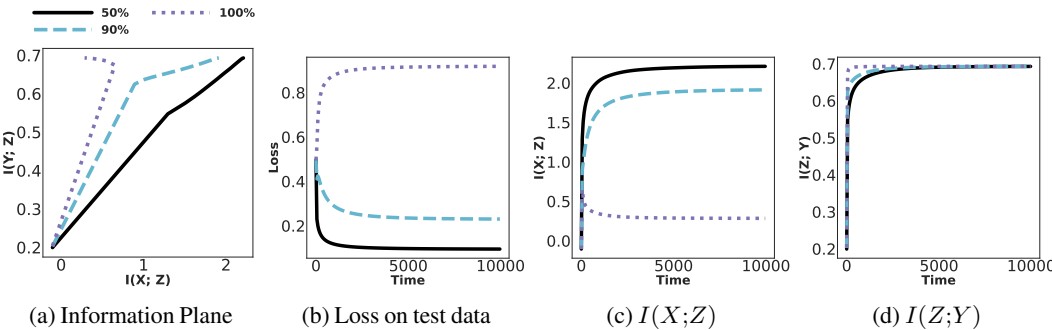

Figure 4: Effect of shortcut effectiveness on the MI trajectory. Shortcut is added to different percentages of even images in each experiment. MI $I(X;Z)$ decreases with shortcut prevalence. Animated GIF of the plot with more degree of variation of shortcut effectiveness can be viewed here. Quantities plotted for subplots (a–d) are consistent with Figure 3.

**MNIST with synthetic shortcut.** We train a model for the binary classification task of classifying MNIST images into odd or even digits. We add a small white patch on one corner of *all even* digits of the MNIST training dataset as a shortcut (Figure 1). The model can use the shortcut patch alone to accurately classify the images into odd and even. We compare the MI during the training evolution on datasets with and without shortcuts in Figure 3.

In Figure 3a and Figure 3c, we observe that the mutual information $I(X;Z)$ on the test data increases initially during training but then the network latches onto the shortcuts, after which the mutual information decreases sharply. It can also be observed in Figure 3b that generalization error increases after the point at which MI starts to decrease, indicating that the network explores the more optimal region of the solution space in the initial training epochs before discovering shortcuts. This is consistent with the findings of Shwartz-Ziv & Tishby (2017) on the behaviour of SGD: in the initial phase, SGD explores the multidimensional space of solutions. When it begins converging, it arrives at the diffusion phase in which the network learns to compress (Shwartz-Ziv & Tishby, 2017). In both settings, the model achieves high training accuracy, i.e., $I(Z;Y)$ (Figure 3d) but the difference in test set loss is striking (Figure 3b).

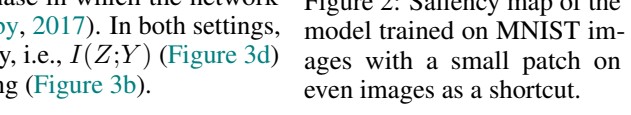

Figure 2: Saliency map of the model trained on MNIST images with a small patch on even images as a shortcut.

**Visualisation.** To visually verify that the network is learning the shortcut in the above experiment, we generate a saliency map. Using

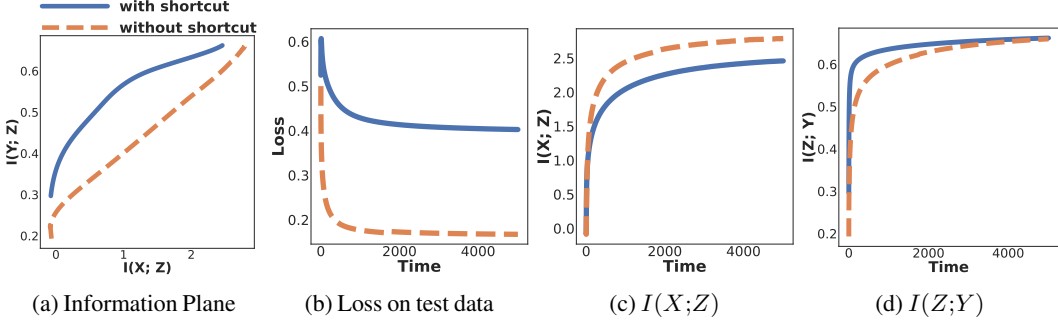

(a) Information Plane      (b) Loss on test data      (c) $I(X;Z)$      (d) $I(Z;Y)$

Figure 5: Mutual information profile for CelebA dataset with hair attribute as a shortcut. Plot of $I(X;Z)$ in (a) and (c) show that shortcuts results in reduced $I(X;Z)$. Animated GIF of the plot can be viewed here.

the finite-difference estimation to find the gradient of the class probability w.r.t. to the input pixels, we verify that the network predominantly uses the shortcut in the image to predict the class (Figure 2).

**Effect of partially correlated shortcuts.** In real-world data, shortcuts are often partially correlated with the output, i.e., the model cannot classify with $100\%$ accuracy using only the shortcuts. To understand the effect of partially correlated shortcuts on the training dynamics, we construct different datasets with varying degrees of shortcut efficacy. Instead of corrupting all the even images, we add a small white patch on one corner only to a specific percentage ($50\%$–$100\%$) of training images of even digits. We plot the MI for varying degrees of corruption in Figure 4 for 1000 training points sampled uniformly.

We observe that as the effectiveness of the shortcut increases, $I(X;Z)$ on the OOD test data converges to a lower value indicating the ability to perform more compression. We also note an interesting behaviour during the training evolution: when the shortcut is partially correlated, MI does not decrease significantly as compared to the $100\%$ effective shortcut as shown in Figure 4c. We speculate that, while in these cases (e.g. $80$–$90\%$ shortcut efficacy), the model is able to recover some generalization ability, its ability to discover high generalization solutions is irrevocably deteriorated once it discovers the minima corresponding to the shortcut solution. This is different than the observation made by Kirichenko et al. (2022) that fine-tuning only the last layer of the model is sufficient to significantly reduce the impact of spurious features and improve worst-group-performance of the models. Lubana et al. (2023) also studied this problem in terms of linear mode-connectivity and also arrived at a similar conclusion as we suggest; namely that naive fine-tuning can fail to eliminate spurious correlations in the dataset.

**CelebA with natural shortcuts.** We also test our hypothesis on a dataset containing natural shortcuts. We curate images from the CelebA dataset such that all training images tagged as male have the "black hair color" attribute, while images tagged as female have the "blonde hair color" attribute. We train the network to classify facial images into the male and female categories[2], while the network may use hair color as a shortcut for accurately classifying the training images, it fails to generalize on the OOD test data. Since the dataset is not controlled, some images from both classes have a black color in different parts of the image (background, clothes, etc.), reducing the effectiveness of the hair color attribute. The test data contains no correlation between hair colour and gender label. We plot the $I(X;Z)$ trajectory (calculated on the OOD test data) of the model trained on data with and without shortcuts for 100 sample training points sampled uniformly in $\log$ scale (Figure 5).

We observe a MI profile similar to Figure 4. On data without the shortcut, MI increases consistently, while in the presence of shortcuts, MI converges to a lower value — suggesting the network learns less information about the input space to solve the classification task when the training data contains shortcuts. This validates our hypothesis on real-world data with natural shortcuts.

---

[2]The CelebA dataset only provides binary labels and we do not know how the gender attribute was assigned. Therefore it should be considered as nothing more than an arbitrary class in this experiment.

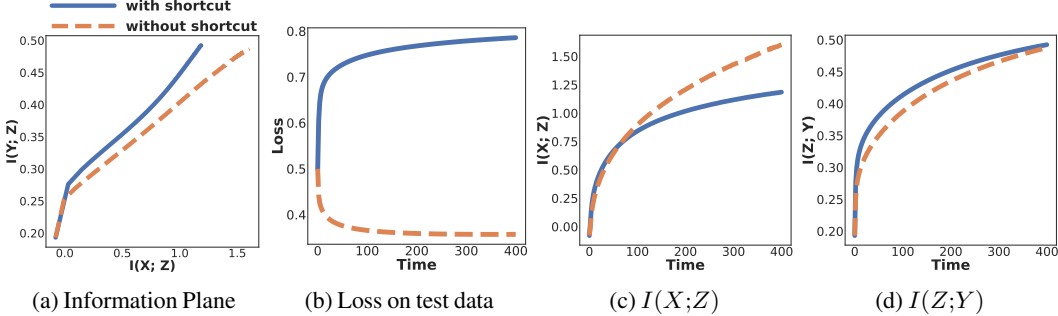

(a) Information Plane  (b) Loss on test data  (c) $I(X;Z)$  (d) $I(Z;Y)$

Figure 6: Mutual information profile for the Waterbird dataset with background context as shortcuts. Plot of $I(X;Z)$ in (a) and (c) show that shortcuts results in reduced $I(X;Z)$. Animated GIF of the plot can be viewed here.

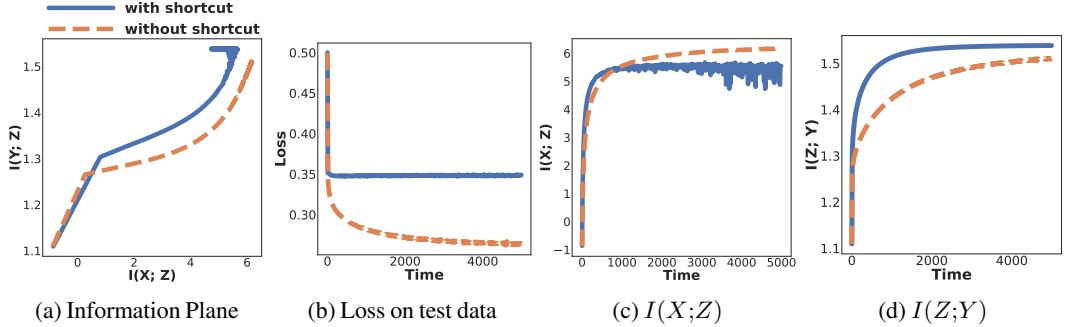

(a) Information Plane  (b) Loss on test data  (c) $I(X;Z)$  (d) $I(Z;Y)$

Figure 7: Mutual information profile for the NICO dataset with background context as shortcuts. Plot of $I(X;Z)$ in (a) and (c) show that shortcuts results in reduced $I(X;Z)$. Animated GIF of the plot can be viewed here.

**Waterbird Dataset.** Vision models often rely on background context for learning discriminatory features, which may not be present during the real-world data, resulting in a decrease in test time performance (Ribeiro et al., 2016b). Since the background varies in different images, such shortcuts are more complex — it is more challenging to inspect datasets visually compared to the shortcuts in the MNIST or CelebA datasets. To simulate such a scenario, we use the Waterbird dataset (Sagawa et al., 2020) to test our proposed method on datasets with background contexts as shortcuts. The Waterbird dataset (Sagawa et al., 2020) combines images from the Caltech-UCSD Birds dataset (Wah et al., 2011) and with image backgrounds from the Places dataset (Zhou et al., 2016). The dataset is divided into two classes, waterbird and landbird, sampled such that training waterbird images from the CUB dataset have water background from the Places dataset and landbird images from the CUB dataset have the land background context from the Places dataset, creating a spurious correlation with the background and the target classes. However, the test data show no correlation between the background and the target classes. It has been found that the models trained on the waterbird dataset rely on the background context and achieve high training accuracy (97.1%), but fail to generalize on the test images, achieving only 71.0% accuracy (Sagawa et al., 2020; Nguyen et al., 2023; Liu et al., 2021a; Setlur et al., 2023).

We plot the MI values during the training evolution as shown Figure 6 and compare MI values when model is trained on data with/without background context as shortcut. We calculate $I(X;Z)$ on the OOD test data that doesn't contain correlation between class labels and the background. We can observe in Figure 6c that mutual information $I(X;Z)$ convergences to a lower value when the training images contain background as spurious correlation, validating that our method can be used on datasets with more complex shortcuts and mutual information $I(X;Z)$ for automatic detection of shortcuts which can lead to decrease in generalization at test time.

**NICO Dataset.** Most real-world classification tasks are multi-class learning problems, unlike the datasets discussed so far. To validate our proposed method on multi-class classification, we use the

| | MNIST | CelebA | Waterbird | NICO |
|---|---|---|---|---|
| Energy | ✗ | ✗ | ✓ | ✓ |
| Entropy | ✗ | ✗ | ✗ | ✓ |
| ODIN | ✗ | ✗ | ✗ | ✓ |
| Mahalanobis | ✓ | ✓ | ✗ | ✓ |
| MaxLogit | ✗ | ✗ | ✓ | ✓ |
| MaxSoftmax | ✗ | ✗ | ✗ | ✓ |
| MCD | ✗ | ✗ | ✗ | ✗ |
| **Ours** | ✓ | ✓ | ✓ | ✓ |

Table 1: Comparison against OOD detectors. Symbols (✓, ✗) denote that if the method can detect shorcuts. Baseline methods are unable to detect shortcuts for all the datasets. In contrast, our MI-based detector can detect shortcuts for all the datasets. We used $\tau = 0.90$ to threshold the FPR@95TPR; FPR@95TPR and AUROC values can be found in Table 2.

Non-I.I.D. Image with Contexts (NICO) dataset (He et al., 2021). The NICO dataset is specifically designed for benchmarking out-of-distribution generalization. It simulates a real-world setting where the test distribution induces an arbitrary shift from the training distribution, which violates the traditional I.I.D. hypothesis of most ML methods. NICO contains 19 classes and 188 contexts, with class and background context labels available for all images. We sample five classes from the NICO dataset: bear, cow, giraffe, hot air balloon and tent, such that each class has background context as a shortcut (cow - grass, bear - water, giraffe - night/dim lighting, hot air balloon - outdoor, tent - autumn scene). Since some of the background images are similar, e.g., water appears in both images tagged as 'water' background, and some of the outdoor background context images, shortcuts are not 100% effective as in the MNIST experiment (Figure 4). Test images are sampled randomly, so there is no spurious correlation between the target class and the background context. Models relying on background context for learning discriminatory feature fails to generalize on the test dataset. Murali et al. (2023) showed that for cow vs. bird, classification accuracy drops from 97.2% on training images to 59.9% on test images (with different background context distribution).

We plot the MI values during the training evolution as shown in Figure 7. Similar to previous experiments, mutual information $I(X;Z)$ calculated on the test data converges to a lower value when the training data contains shortcuts, thus validating that our method can be used in more general multi-class settings and MI values are suggestive of shortcuts present in the training dataset.

**Baselines.** In contrast to prior works, such as (Müller et al., 2023) that have proposed methods based on domain knowledge or a human-in-the-loop approach, our method is domain-agnostic, i.e., it does not require any domain knowledge or human annotation. Thus, **a direct comparison of these methods would be misleading**. Instead, we benchmark our method against OOD detection methods to demonstrate the efficacy of our method in detecting shortcuts compared to the existing OOD detection methods. We use two metrics from the OOD literature — Area under the Receiver Operating Characteristic (AUROC) and FPR@95TPR, which refers to the false positive rate at a $95\%$ true positive rate. The smaller the FPR@95TPR, the better the OOD discrimination performance (Bitterwolf et al., 2022). To detect distribution shifts, a threshold ($\tau$) is applied to the FPR@95TPR values. We compare against popular OOD detectors — Mahalanobis (Lee et al., 2018b), Monte Carlo Dropout (MCD) (Gal & Ghahramani, 2016), Energy-based (Liu et al., 2021b), ODIN (Liang et al., 2020), Entropy-based (Macedo et al., 2022), MaxLogit (Hendrycks et al., 2022), MaxSoftmax (Hendrycks & Gimpel, 2018). We use PyTorch-OOD library (Kirchheim et al., 2022) to implement the OOD baselines and train the models with cross-entropy and Centre Loss (Wen et al., 2016). As shown in Table 1, OOD detectors fail to detect shortcuts/spurious correlation. Ming et al. (2021) and Zhang & Ranganath (2023) also found OOD detectors ineffective for detecting spurious correlations.

## 5 Probing medical datasets for shortcuts

Deploying machine learning models in clinical settings has been challenging (Ghassemi et al., 2019; Kelly et al., 2019) as healthcare data often have shortcuts (DeGrave et al., 2021; Nauta et al., 2021).

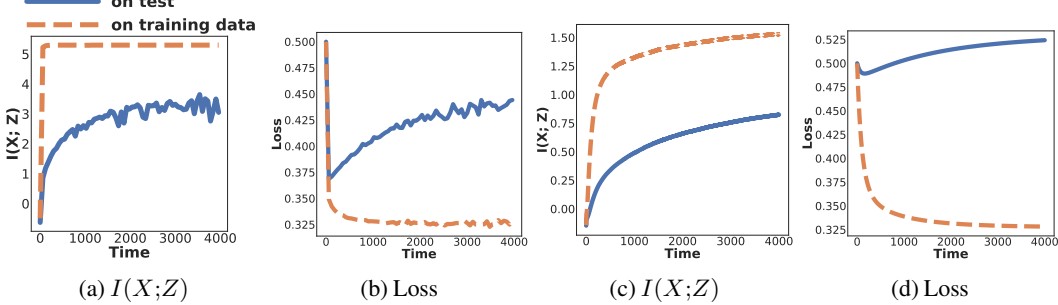

(a) $I(X;Z)$      (b) Loss      (c) $I(X;Z)$      (d) Loss

Figure 8: (a) Mutual information profiles on the diabetic retinopathy dataset. (b) Training and test loss during the training evolution. (c) Mutual information profiles on the breast cancer classification dataset. (d) Training and test loss during the training evolution on breast cancer classification dataset. Mutual information $I(X;Z)$ for the out-of-distribution test data converges to a lower value and thus can be used to detect shortcuts and distribution shifts between the training and test data. Plots of loss on training and test data shows that model doesn't generalize well on the test dataset.

Unlike MNIST, CelebA, Waterbird and NICO datasets, where shortcuts were either synthetically added or the dataset was sampled to introduce known shortcuts in the datasets, detecting unknown shortcuts manually in medical datasets is non-trivial. To verify if our proposed method can detect shortcuts on a more realistic dataset when the shortcuts are not precisely known beforehand, we use the recently proposed BenchMD benchmark (Wantlin et al., 2023). The BenchMD dataset is designed to evaluate performance on out-of-distribution data from different hospitals, representing naturally-occurring distribution shifts that frequently degrade the performance of models trained on medical/healthcare datasets. For example, Wantlin et al. (2023) noted that for diabetic retinopathy grading classification, when the model is evaluated on images from a different hospital, model accuracy decreases. We used the same training (Messidor dataset) and test data (APTOS 2019 dataset) for the diabetic retinopathy classification dataset from BenchMD. We compute mutual information $I(X;Z)$ for the out-of-distribution APTOS dataset (Karthik, 2019) using the model trained on the Messidor dataset (Abràmoff et al., 2013). We observe mutual information $I(X;Z)$ on the APTOS dataset is lower compared to the Messidor dataset, suggesting that the Messidor dataset contains shortcuts that do not generalize well to out-of-distribution data. Loss in Figure 8b and Figure 8d is calculated on the held-out samples from the training data and test data which verifies that model indeed does not generalize well on the test data.

We also experiment with the breast cancer classification datasets, BreakHis dataset (Spanhol et al., 2016) and the BACH dataset (Polónia et al., 2019) and study if the mutual information can be used to detect distribution shift between the BreakHis and BACH dataset. We train the model on the BreakHis dataset and similarly compare the mutual information $I(X;Z)$ on the held-out training data and test data (BACH dataset). As expected, the mutual information $I(X;Z)$ value for the BACH is lower, suggesting shortcuts in the BreakHis dataset.

## 6 DISCUSSION

In this work, we sought to understand why neural networks tend to learn shortcuts through the lens of the information bottleneck. We demonstrated across multiple datasets empirical support for our hypothesis that MI can be used as a domain-agnostic tool for the automatic detection of dataset shortcuts. This is in contrast to many interpretable machine learning methods that often require domain knowledge. Since computing MI $I(Z;X)$ on the OOD data doesn't require labels, *shortcuts can be detected without labeled data*. Our method is easy to implement with low computational cost as NTK is a kernel-based and does not need to be trained using gradient descent. Our method can be used in complementary with existing OOD detectors to detect all types of distribution shifts, including shortcuts which are often not detected by existing OOD detectors. We used the NTK to estimate MI, limiting our approach to infinite-width neural networks. Alternative MI estimation techniques may be required to generalize our method to finite-width networks. One limitation of our method is that it does not work on reverse spurious correlations (Arjovsky et al., 2020).

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

# A APPENDIX

## A.1 ADDITIONAL EXPERIMENT

**CelebA with synthetic shortcut.** We also validate our hypothesis on CelebA dataset (Liu et al., 2015). Similar to the MNIST experiment, we created a synthetic shortcut by adding a small white patch on one corner of training images tagged as *male*. The model is trained for the binary classification of images into CelebA's male and female classes. As shown in Figure 9, the mutual information $I(X;Z)$ on the OOD test data converges to a lower value in the presence of shortcuts in the training data. It can be observed from Figure 9a that the model can achieve higher accuracy on training data even by encoding less information about the input space using shortcuts present in the training data.

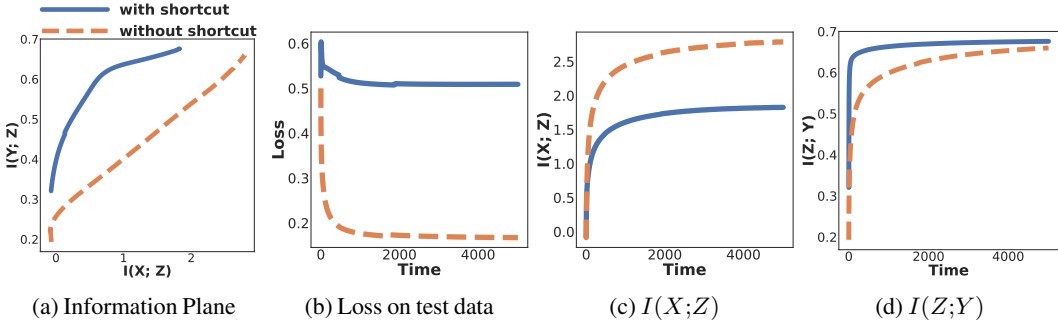

(a) Information Plane   (b) Loss on test data   (c) $I(X;Z)$   (d) $I(Z;Y)$

Figure 9: Mutual information profile for the CelebA dataset with synthetic shortcuts. Plot of $I(X;Z)$ in (a) and (c) show that shortcuts results in reduced $I(X;Z)$. Animated GIF of the plot can be viewed here.

## A.2 IMPLEMENTATION DETAILS

We used the Neural Tangent library (Novak et al., 2019) to compute the NTK and NNGP kernel for a given architecture; JAX (Bradbury et al., 2018) to implement the infinite-width neural network and compute mutual information. We used publicly available datasets for our experiments and sampled the dataset to introduce spurious correlation with the class labels. Code for implementing our method along with the datasets can be found on our **anonymous repository**.

|  | MNIST | | CelebA | | Waterbird | | NICO | |
|---|---|---|---|---|---|---|---|---|
|  | AUROC | FPR | AUROC | FPR | AUROC | FPR | AUROC | FPR |
| Energy | 0.27 | 0.99 | 0.47 | 0.91 | 0.51 | 0.90 | 0.73 | 0.84 |
| Entropy | 0.50 | 1.00 | 0.50 | 1.00 | 0.50 | 1.00 | 0.75 | 0.74 |
| ODIN | 0.50 | 1.00 | 0.50 | 1.00 | 0.50 | 1.00 | 0.73 | 0.79 |
| Mahalanobis | 0.69 | 0.51 | 0.85 | 0.80 | 0.64 | 0.94 | 0.67 | 0.67 |
| MaxLogit | 0.26 | 0.99 | 0.47 | 0.91 | 0.51 | 0.90 | 0.75 | 0.81 |
| MaxSoftmax | 0.50 | 1.00 | 0.50 | 1.00 | 0.50 | 1.00 | 0.76 | 0.75 |
| MCD | 0.50 | 1.00 | 0.50 | 1.00 | 0.50 | 1.00 | 0.50 | 1.00 |

Table 2: AUROC and FPR@95TPR (denoted by FPR in the table) values of OOD detectors on different datasets. While Mahalanobis can detect shortcuts in MNIST, CelebA and NICO, it fails to detect shortcuts in the waterbird dataset. We used $\tau = 0.90$ to threshold FPR values.

## A.3 BASELINES

We benchmark against the following OOD baselines to show our method can detect shortcuts while the existing OOD detectors cannot:

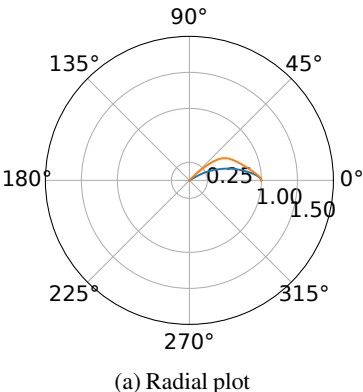

(a) Radial plot

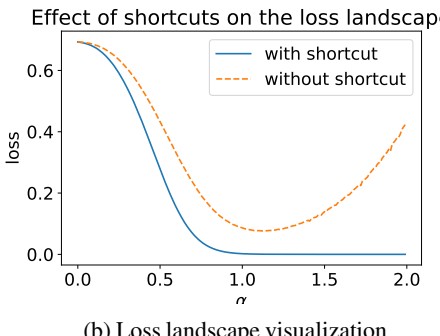

(b) Loss landscape visualization

Figure 10: Visualization of loss landscape on MNIST dataset. (a). polar coordinates $(r_t, \phi_t)$ measuring the deviation from the linear path between initialisation and converged parameters in the weight space during the optimization. (b). 1-D visualization of loss landscape.

**Energy-based OOD detector.** Liu et al. (2021b) proposed an energy-based method to detect OOD inputs using an energy score. The model maps the input to a single, non-probabilistic scalar called the energy. The method uses energy instead of softmax for calculating the confidence scores. Data samples with high energy are considered as OOD inputs and vice versa.

**Entropy-based OOD detector.** Macedo et al. (2022) introduced IsoMax loss to train the model, which improves the OOD detection to tackle the overconfidence of SoftMax loss. IsoMax loss force the logits to depend only on the distances from the high-level features to the class prototypes. Let $f_\theta(x)$ represent the feature embeddings for the input $x$, $p_\theta^j$ represent the learnable prototype with class $j$, and $y^k$ represent the label of the correct class; IsoMax loss can be described as following:

$$L_I(y^k|x) = -\log \frac{\exp(-d(f_{\theta(x)}, p_\theta^k))}{\sum_j \exp(-d(f_{\theta(x)}, p_\theta^j))} \tag{10}$$

**Monte Carlo Dropout (MCD).** Gal & Ghahramani (2016) introduced MCD, which uses Dropout (Srivastava et al., 2014) as an Bayesian approximation to the Gaussian Processes. MCD uses variance of the output probabilistic distribution to estimate the model's confidence and detect OOD samples.

**Mahalanobis.** Lee et al. (2018b) proposed to measure the probability density of test samples in feature spaces using class-conditional Gaussian distribution. They defined the confidence score using *Mahalanobis* distance with respect to the closest class-conditional distribution, where its parameters are chosen as empirical class means and tied to the empirical covariance of training samples.

**MaxSoftmax.** Hendrycks & Gimpel (2018) observed that correctly classified examples tend to have greater maximum softmax probabilities than incorrectly classified and OOD. They showed that the prediction probability of OOD samples is lower than the prediction probability of in-distribution samples, and thus, observing prediction probability statistics can help in detecting OOD samples.

**MaxLogit.** Hendrycks et al. (2022) proposed to use the negative of the maximum unnormalized logit for an anomaly score $-\max_k f(x)_k$, which they call *MaxLogit* as a confidence score for detecting OOD samples.

**ODIN.** Liang et al. (2020) proposed a simple change to softmax to improve OOD detection. ODIN used a temperature scaling in the softmax and adds small perturbations to the training inputs for more effective OOD detection.

## A.4 Effect of shortcut on the loss landscape:

We visualize the loss landscape of neural networks to understand the effect of shortcuts on the optimization trajectory. We plot loss along a linear path connecting the initial parameter $\theta_o$ and converged parameter $\theta^*$ in the weight space (Goodfellow et al., 2014) and polar coordinates $(r_t, \phi_t)$ plot measuring the deviation from the linear line between $\theta_i$ and $\theta^*$ (Figure 10). We parameterize the line with $\alpha$ such that $\theta = (1-\alpha)\theta_i + \alpha\theta^*$. Polar coordinates can be calculated using $r_t = \frac{||\triangle\theta_t||}{||\triangle\theta_o||}$ and $\phi_t = \arccos\frac{\triangle\theta_t \times \triangle\theta_o}{||\triangle\theta_t|| \times ||\triangle\theta_o||}$, where $\triangle\theta_t = \theta_t - \theta^*$. We observe that the loss landscape around $\theta^*$ in the case of shortcuts is surprisingly flat as compared to the valley-like shape for a model trained on data not containing shortcuts using the MNIST dataset. The polar plot shows that the optimizer deviates less from the linear trajectory when trained with shortcuts.

## A.5 Additional Comments

We chose to use NTK to calculate mutual information, as in contrast to other methods such as MINE (Belghazi et al., 2021), NTK doesn't require training using gradient descent due to its kernel behaviour at the infinite limit. NTK can give the mutual information profile during the entire training evolution without much computation overhead, whereas MINE and other neural network-based methods need to be trained for hundreds of epochs to approximate the MI between the two variables, which would be feasible and computationally expensive to calculate MI for every epoch during the training evolution. To summarise, our work is different from existing shortcut detection methods in the literature as the proposed method doesn't require any human annotation or human-in-the-loop to detect shortcuts, i.e., our method is domain agnostic and does not require human expertise to detect shortcuts. Moreover, due to the kernel behaviour of NTK, mutual information can be computed for the entire training evolution without much computational overhead. Our method can be used to check for shortcuts in the training data before deploying the model on new unlabelled test data. For e.g., in the medical dataset experiment, we trained the model on the Messidor dataset and used APTOS dataset as a test dataset. Our method was able to detect shortcuts in the context of the APTOS and Messidor dataset (Figure 9), which is in line with the recent benchmarking result on these two datasets, i.e., models trained on APTOS do not generalize well [3]. We believe this is quite a useful application of our method, especially in domains where it is difficult to detect shortcuts manually.

