# OpenReview forum: "Detecting Shortcuts using Mutual Information"
_ICLR.cc/2024/Conference — Submitted to ICLR 2024_

### Official Review · Reviewer_XNRX · 2023-10-29

**Soundness:** 2 fair
**Presentation:** 2 fair
**Contribution:** 2 fair
**Rating:** 3
**Confidence:** 4

**Summary:**

This paper propose to detect shortcut by evaluating the mutual information between the latent variable Z of a network and the input X. The intuition is that the latent variable Z of a network learning shortcuts would have a lower mutual information with the input X. Based on the intuition, the paper propose to detect shortcut by calculating mutual information using neural tangent kernel (NTK).

**Strengths:**

1). Using mutual information to detect whether networks is an interesting topic.

2). This paper provide a detailed related work introduction.

**Weaknesses:**

The proposed method seems infeasible.

1). In algorithm 1, a dataset is said to contain shortcut if $I(X_{test};Z)<I(X_{tr};Z)$. However, since $X_{test}$ have different distribution than $X_{tr}$, it seems natural to have $I(X_{test};Z)<I(X_{tr};Z)$. Therefore an important question need to be answered: are there any datasets satisfying $I(X_{test};Z)\geq I(X_{tr};Z)$?  There lacks empirical evidences in this paper to answer the question and I do not think the algorithm 1 could effectively detect shortcuts.

2). Unlike the proposed algorithm 1, the experiments in this paper, on the other hand, mainly compare networks trained on two datasets "with" and "without" shortcut. This approach also have problems since it requires comparing with a network trained on dataset that is "without" shortcut. By defining "without shortcut", it also involves domain knowledge and human expertise to detect shortcuts.

Based on the above two points, I think that the propose method have major flaws.

Minor issues:

1). the introduction of estimating mutual information using NTK is vague. For example, what is the definition of $\Theta(x,X)$? $\sigma$ is used in Eq.5 but it is written as $\Sigma$ in Eq.7.

2). Figure legends (e.g. Fig.4) could be more detailed.

**Questions:**

As mentioned in the weakness section, I have concerns over the propose method.

1). For those datasets without shortcut, are they satisfy $I(X_{test};Z)=I(X_{tr};Z)$? Please provide empirical evidences to show that algorithm 1 is feasible.

2). I notice that at the begining of the training, before the mutual information $I(X;Z)$ starts to be different between "with shortcut" and "without shortcut", the test loss has become different (e.g. 100 epoch in Fig.6 and 1000 epoch in Fig.7).  Why is it?

---

> ### Author Response · Authors · 2023-11-18
> **Rebuttal**
>
> Thank you for the detailed review and insightful feedback. Here is the further explanation/answers to questions raised in the review:
>
> > Are there any datasets satisfying $I(X_{test}; Z) \geq I(X_{tr}; Z)$? There lacks empirical evidences in this paper to answer the question and I do not think the algorithm 1 could effectively detect shortcuts.
>
> While $I(X_{test}; Z)$ cannot be greater than $I(X_{tr}; Z)$, however, for $X_{test}$ sampled from same distribution as $X_{tr}$, $I(X_{test}; Z) \approx I(X_{tr}; Z)$, as shown in the different experiments.
>
> > Unlike the proposed algorithm 1, the experiments in this paper, on the other hand, mainly compare networks trained on two datasets "with" and "without" shortcuts. This approach also has problems since it requires comparing with a network trained on dataset that is "without" shortcut. By defining "without shortcuts", also involves domain knowledge and human expertise to detect shortcuts.
>
> We use `with' and 'without shortcut' to denote shortcuts in the controlled experiments to demonstrate that mutual information can be used to distinguish between the two datasets. We demonstrated the practical use case in Section 5 where we sample $X_{tr}$  and $X_{test}$ from two different medical imaging datasets. Our method can detect the presence of shortcuts, which is in fact corroborated when the model train on $X_{tr}$ is evaluated on $X_{test}$.
>
> >For those datasets without shortcut, are they satisfy  $I(X_{test}; Z) = I(X_{tr}; Z)$? Please provide empirical evidences to show that algorithm 1 is feasible.
>
> $I(X_{test}; Z) and I(X_{tr}; Z)$ will similar when sampled from the same distribution. In the experiments reported, MI $ I(X_{tr}; Z) $ is calculated on different samples from the training distribution.
>
> >  I notice that at the begining of the training, before the mutual information  starts to be different between "with shortcut" and "without shortcut", the test loss has become different (e.g. 100 epoch in Fig.6 and 1000 epoch in Fig.7). Why is it?
>
> Our intuition in the initial phase network explores the more optimal region of the solution space in the initial training epochs before discovering shortcuts. This is consistent with the findings of Shwartz-Ziv & Tishby (2017) on the behaviour of SGD.
>
>
> [1] Ravid Shwartz-Ziv and Naftali Tishby. Opening the black box of deep neural networks via information.
> 2017

---

### Official Review · Reviewer_8GQW · 2023-10-31

**Soundness:** 2 fair
**Presentation:** 2 fair
**Contribution:** 3 good
**Rating:** 3
**Confidence:** 4

**Summary:**

The author claims that the failure of deep neural networks to generalize to out-of-distribution (OOD) data is often caused by shortcuts or spurious correlations in the training data, leading to a performance drop due to distribution shift. This paper aims to understand shortcut learning using information-theoretic tools and propose using mutual information (MI) between the learned representation and the input space as a domain-agnostic metric for detecting shortcuts in training datasets. To study the training dynamics of shortcut learning, a framework based on the Neural Tangent Kernel (NTK) in introduced, able to detect shortcuts and spurious correlations in training data without requiring class labels for the test data. Empirical experiments on multiple datasets, including MNIST, CelebA, NICO, Waterbirds, and BenchMD, demonstrate that MI can effectively detect shortcuts.

**Strengths:**

1. This paper can be viewed as a complementary method to existing out-of-distribution (OOD) detectors. It offers a domain-agnostic metric to detect shortcuts and provides a way to diversify the training data before deploying a model in safety-critical domains. By identifying shortcuts and spurious correlations in training datasets, it helps improve the robustness and reliability of models when faced with distribution shifts, making it a valuable tool for ensuring model performance in real-world applications.

2. The quality of the writing is commendable, and the experiments conducted to establish the effectiveness are adequate.

**Weaknesses:**

1. The core concept underpinning the method lacks novelty. In essence, during the training phase, the objective is to minimize $I(X; Z)$ on the training data $D_tr$. However, it becomes evident during the testing phase that $I(X_{te}; Z) < I(X_{tr}; Z)$, provided a domain gap exists between the training data $D_{tr}$ and the test data $D_{te}$. Subsequently, equations (4) - (10) are all employed to compute $I(X; Z)$ using well-established formulations from prior literature. Hence, the methods devised are not restricted to the detection of shortcuts and can be applied to data exhibiting substantial domain gaps. I recommend that the authors conduct additional experiments on out-of-distribution (OOD) detection.

2. Based on the experiments on partially correlated shortcuts as depicted in Figure 4, I have reservations about the method's limitations when it comes to detecting datasets with partially correlated shortcuts. This is because the hard line in Figure 4(a) appears to be very close to the dotted line in Figure 3(a). I recommend that the authors conduct additional experiments to further investigate the detection of datasets with partially correlated shortcuts.

**Questions:**

See weaknesses as above.

---

> ### Author Response · Authors · 2023-11-18
> **Reply to 8GQW**
>
> Thanks for the detailed feedback! We will run additional experiments on OOD for future submission.

---

### Official Review · Reviewer_XhXU · 2023-11-02

**Soundness:** 1 poor
**Presentation:** 1 poor
**Contribution:** 2 fair
**Rating:** 3
**Confidence:** 3

**Summary:**

The authors utilize a neural tangent kernel framework to approximate the mutual information between input and learned representations. They hypothesize that mutual information should be smaller in the test set compared to the training set if there's a distribution shift present. They test their ideas on a number of datasets and compare their proposed methodology with out-of-distribution (OOD) detection algorithms.

**Strengths:**

- The authors' topic of choice is timely, and is of both theoretical and practical significance.
- The authors involve a number of datasets in their experiments, and go beyond frequently used (semi)synthetic datasets in their investigation. Domain-specific empirical examination is especially likely to be informative in this topic.

**Weaknesses:**

I believe that the issue the authors address requires conceptual clarity in terms of defining and diagnosing the problem, and a carefully constructed methodology and accompanying experiments to support the arguments made. I believe that the paper has important shortcomings in these respects. I summarize my main criticisms below, and provide detailed questions and observations regarding these in the next section.
- The paper presents no systematic justification of their proposed method, apart from some conjectural statements.
- The paper is imprecise, unclear, and/or inconsistent with their conceptual structure or central notation.
- The experiments do not seem consistent with the onus of the paper's argumentation, present insufficient details on the specifics, and do not pursue some critical avenues to their fullest extent.

**Questions:**

## On justification of the proposed method:
- I do not think it is clear why the paper's method of choice (i.e. comparing the mutual information between input and learned representations) should be a good choice for detecting this phenomenon. An example:
	- Let's assume a binary label $Y$ and a binary feature $S$, and that $p_{train}(S=1|Y=1) = p_{train}(S=0|Y=0) = 1$, and  $p_{test}(S=1|Y=1) = p_{test}(S=0|Y=0) = 0.5$, which conforms to the Eq's 2 and 3. Also let $p_{train}(Y = 1) = p_{train}(Y = 1) = 0.5$. $Y$ can be odd/even label and $S$ can be the presence of the white patch.
	- Then, according to paper's arguments a trained model would learn to reduce the representation of $X$ to the presence of the white patch $g(X) = Z = S$ to maximally compress $X$. In the test set the network would still extract the presence of the white patch from the OOD samples. Why would then $I(X;Z)$ be different under $p_{train}$ and $p_{test}$?
- Moreover, can the authors definitively claim that $I(X_{test}; Z) < I(X_{tr};Z)$ can only be due to the presence of spurious correlations? If not, how reliable should this method be considered for detecting shortcut learning?
- The suggested methodology might make sense under some distribution shifts, but it is authors' responsibility to describe and explain this while presenting their method.

## On conceptual clarity and notation consistency
- The authors do not make clear notational distinctions between random variables and specific values they can take, and use inconsistent notation seemingly without explanation:
	- After introducing $X$ and $Y$ as random variables, why do we revert to $x$ and $y$ in Eq.s 2 and 3?
	- On Eq.s 8 and 9 why do we have e.g. $X$ and $x$ in the same equation?
	- Why is $\mathcal{Y}$ never explicitly introduced?
	- Do $p$ and $P$ refer to different mathematical objects? If so, why are they not explicitly introduced?
	- Why is $s$ always lower case?
	- Why do we have $X_{tr}$ but not $Z_{tr}$?
- The generative model implied in Section 3 is not clear, and it's not clear how it relates to distribution shifts in question.
- Definition 2 is unclear. What is $\Gamma$? Is it a function that outputs scalar values, such that we can have order relationships?
- Proposition 1 is not a provable mathematical statement, so it should be named a conjecture or a hypothesis. Even as a hypothesis it is imprecisely stated.
- Pg. 1: "A shortcut is a distribution shift..." I think defining shortcut as a type of distribution shift is both unhelpful and is inconsistent with the rest of the literature.

## On experiments
- Both in the abstract and the introduction, as well as in the Algorithm 1, the authors propose examining $I(X_{test}; Z) < I(X_{tr};Z)$ as a way to determine presence of shortcut learning, yet in none of the experiments they apply this methodology to decide on this, until Section 5. Neither is this information present in any of their figures before Figure 8.
- Although deferring some details to the supplementary material is understandable, the authors include no details whatsoever on their experiment setting. This ranges from the used model families to how $Z$ was obtained.
- The comparison with OOD methods (i.e. how the baselines were used for this task) should be introduced in a more detailed fashion.
- Figure 2 is a singular demonstration of what the model learns in the presence of strong spurious correlations. A method for quantifying this is needed, in a way that generalizes to other experiments as well.

---

### Official Review · Reviewer_cP5y · 2023-11-08

**Soundness:** 2 fair
**Presentation:** 3 good
**Contribution:** 2 fair
**Rating:** 3
**Confidence:** 4

**Summary:**

This paper proposed a framework for detecting spurious correlations or shortcuts implied in training datasets. The main hypothesis of this paper is that the information between input and embedding would be low provided that there are shortcuts in a dataset. The author leveraged a neural tangent kernel to estimate mutual information between input and embedding representation and empirically represented that their hypothesis is valid on the synthetic (MNIST with shortcuts), benchmarks (Waterbirds, CelebA, and NICO), and real-world medical datasets.

**Strengths:**

- The empirical results support their hypothesis that the $I(X;Z) of a dataset with a shortcut is lower than that without a shortcut.
- The method is simple and easy to follow.

**Weaknesses:**

- The proposed method is limited in the real-world application scenario. The proposed method requires a 'without shortcuts dataset' to detect whether there are shortcuts in the training dataset. I am not sure how many cases we can prepare the 'without shortcuts dataset' before we know whether the training dataset has a shortcut.
- (Kirichenko et al., 2022) represented that the model trained on Waterbirds using ERM has the ability to classify the foreground-only and background-only datasets. It conflicts with the main hypothesis that the model trained with the shortcut dataset will have a low $I(X;Z).
- The experiment graphs show that the mutual information is less discriminative than the losses, which diminishes the necessity of using mutual information to detect the existence of shortcuts in the training dataset.

Minor corrections
- Page 5 MNIST with synthetic shortcut) Figure 1 -> Figure 2
- Figure 2) Please denote the used Saliency map.
- Definition 2) Please state that the higher $\Gamma$, the better the generalization.

**Questions:**

- The proposed algorithm and the experiment setting are different. If I denote an original training and test dataset as $D_{tr}$ and $D_{te}$, respectively, and the shortcut added training dataset as $D_{tr}^{sc}$, then the Figure 3(c) plots '$I(X_{test};Z)$ trained on $D_{tr}$' and '$I(X_{test};Z)$ trained on $D_{tr}^{sc}$'. However, the algorithm seems to be denoted to compare '$I(X_{test};Z)$ trained on $D_{tr}$' and '$I(X_{tr};Z)$ trained on $D_{tr}$'.
- Algorithm 1 step 1) Why $\mathcal{F}$ is initially required?
- Figure 3 with shortcut line vs Figure 4 100% line) I think they are the same experiment, but the graphs differ.

---

> ### Author Response · Authors · 2023-11-18
> **Rebuttal**
>
> Thank you for the detailed review and feedback!
>
> > The proposed method is limited in the real-world application scenario. The proposed method requires a 'without shortcuts dataset' to detect whether there are shortcuts in the training dataset. I am not sure how many cases we can prepare the 'without shortcuts dataset' before we know whether the training dataset has a shortcut.
>
> We use `with' and 'without shortcut' to denote shortcuts in the controlled experiments to demonstrate that mutual information can be used to distinguish between the two datasets. We demonstrated the practical use case in Section 5 where we sample $X_{tr}$
>  and $X_{test}$  from two different medical imaging datasets. Our method can detect the presence of shortcuts, which is in fact corroborated when the model trained on $X_{tr}$ is evaluated on $X_{test}$.
>
> > (Kirichenko et al., 2022) represented that the model trained on Waterbirds using ERM has the ability to classify the foreground-only and background-only datasets. It conflicts with the main hypothesis that the model trained with the shortcut dataset will have a low $I(X;Z).
>
> Thanks for pointing out this work. One major difference is that we use Neural tangent kernels for training the model and calculating the mutual information value. Kirichenko et al., 2022 used a finite-width neural network in their study.
>
>
> > The experiment graphs show that the mutual information is less discriminative than the losses, which diminishes the necessity of using mutual information to detect the existence of shortcuts in the training dataset.
>
> While using a trained model on new data $X_{test}$, we usually don't have the corresponding label $Y_{test}$. In contrast to the test accuracy, calculating MI $I(X_{test}; Z)$ does not require a labelled dataset and hence our method can be used to detect shortcuts without requiring test labels. We plotted the test accuracy to empirically demonstrate the connection between $I(X_{test}; Z)$ and the test accuracy.
>
>
> > The proposed algorithm and the experiment setting are different.
>
> In our experiments, we calculated both $I(X_{tes}t; Z)$ and $I(X_{tr}; Z)$ using model trained $D_{tr}$
>
>
>
> > Algorithm 1 step 1) Why $\mathcal{F}$ is initially required?
>
> We need a trained model $\mathcal{F}$ to calculate the mutual information.

---

> > ### Comment · Reviewer_cP5y · 2023-11-22
> >
> > I greatly appreciate your response. After considering the opinions of the other reviewers and the author's response, I tend to maintain my original score.

---

### Meta-Review · Area_Chair_Am5V · 2023-12-12

**Metareview:**

This paper presents the presence of shortcuts in machine learning as an out-of-distribution problem and uses mutual information between input and learned representations using a neural tangent kernel to detect them. Experimental results are presented on many benchmarks.

One major conceptual clarification that is missing from the mutual information formulation of this procedure is the lack of distinction between spurious correlations and true inductive biases (or truly informative features). The true inductive biases will be helpful in any distribution, whereas shortcuts might be different in different distributions. The “no shortcut” setup studied in this work is somewhat reductive and not possible to study on a wide variety of settings. In effect, what the paper is proposing as a shortcut detector might simply correspond to an outlier detector.

The presentation of the paper was clear and many empirical studies were presented to verify the hypothesis.

**Justification For Why Not Higher Score:**

Please see weaknesses above; conceptually the method in the paper was somewhat unsatisfactory.

**Justification For Why Not Lower Score:**

N/A

---

### Decision · Program_Chairs · 2024-01-16

Reject